# Readily Available Water Access is Associated with Greater Milk Production in Grazing Dairy Herds

**DOI:** 10.3390/ani9020048

**Published:** 2019-02-05

**Authors:** Ruan R. Daros, José A. Bran, Maria J. Hötzel, Marina A. G. von Keyserlingk

**Affiliations:** 1Animal Welfare Program, Faculty of Land and Food Systems, University of British Columbia, Vancouver, BC V6T 1Z4, Canada; rrdaros@alumni.ubc.ca; 2Laboratório de Etologia Aplicada e Bem-Estar Animal, Departamento de Zootecnia e Desenvolvimento Rural, Universidade Federal de Santa Catarina, Florianópolis 88040-900, Brazil; jose.alfredo@posgrad.ufsc.br (J.A.B.); maria.j.hotzel@ufsc.br (M.J.H.)

**Keywords:** pasture-based systems, animal welfare, lactating cows

## Abstract

**Simple Summary:**

In Santa Catarina, Brazil, most milk is produced on small-scale farms that utilize grazing as the main form of nutrition; however, the farms differ in how they provide water for their herds, with some herds not providing access to drinking water while on pasture and other herds having unrestricted access to water. In this study, we assessed the milk production on farms that differ in the way drinking water is provided to the herd. Herds with unrestricted access to drinking water produced more milk than herds that had restricted access to drinking water, regardless of the main breed of the herd, and amount of concentrate offered per cow per day. Simple changes in water management practices may positively impact milk production.

**Abstract:**

In this cross-sectional study, we measured the association between water provision and milk production on intensively managed small-scale grazing dairy herds. Farms (*n* = 53) were categorized according to water provision as follows: (1) Restricted—cows did not have access to a water trough while on pasture; and (2) Unrestricted—cows had free access to a water trough while on pasture. Herd main breed and feeding practices were included in a model to assess the effect of water provision category on farm average milk yield/cow/d. The effect of pasture condition and environmental variables on milk production were also assessed, however were not retained on the final model. Herds provided with unrestricted access to drinking water produced on average 1.7 L more milk per cow/d (*p* = 0.03) than herds with restricted access to drinking water. Predominantly Holstein herds produced 2.8 L more milk per cow/d (*p* < 0.01) than non-Holstein herds. Each extra kg of concentrate offered per day increased milk yield by 1.1 L/cow/d (*p* < 0.01). In conclusion, providing free access to drinking water while grazing was associated with greater milk production.

## 1. Introduction

Globally, there are a variety of dairy production systems that directly impact dairy cattle welfare and production [1,2]. Intensively managed grazing systems are widely adopted in areas where pasture growth occurs year-round (e.g., Brazil: [3]; Argentina: [4]; Ireland: [5]; and New Zealand: [6]). These grazing systems often use some form of rotational grazing using a series of paddocks to control grazing, improving forage production and grazing efficiency (see PRV—Voisin Rational Grazing—in South America [7] and MIG—management-intensive grazing in North America [8]). 

To efficiently control grazing, cows are often prevented from leaving the paddock, i.e., they remain in the same paddock until the stockperson decides to move them to the next paddock or gather them for the next milking. However, this practice may have negative effects on animal welfare and productivity if, for example, the cows do not have access to drinking water (hereafter referred to simply as water) in the paddock (e.g., [9]). Such practices may cause thirst, an emotional state long regarded as detrimental to animal welfare [10].

Previous research indicates that in intensively managed grazing herds, restricted access to water is common [11,12]. Surprisingly, little research has been done to measure the effects of restricted access to water on milk production in grazing herds (see systematic review [13]). In a recent case study of a Holstein grazing herd in Argentina cows that had access to a water trough while in the grazing paddock produced more milk compared to cows that had to leave the paddock to drink water [14]. Similarly, in an experimental study non-lactating cows drank less water when the water trough was placed outside of the grazing paddock compared to when the water trough was placed inside the grazing paddock, with the effects being greatest for subordinate animals [15]. However, these studies were not designed to measure the effect of restricted water access on milk production [14,15].

Other factors may influence cows’ willingness to access water troughs. For example, the dry matter (DM) content of the feed consumed and generated by the metabolization of nutrients has been shown to be associated with the amount of drinking water consumed [16]. For grazing cows, diet moisture content varies as fresh forages differ in dry matter (DM) depending on their phenological stages—i.e., juvenile plants have lower DM than mature plants. Also, grazing cows have lower free water intake than zero-grazing dairy cows due to their diet’s lower DM content [17] (p. 179). However, restricting water access for grazing dairy cows even if for short periods of times may impact milk production. Low water intake is known to directly reduce DM intake [18] and consequently, reduce milk production.

Likewise, heat stress reduces cows’ DM intake and milk production [19,20]. To assess heat stress the levels of ambient temperature and humidity are combined into a temperature humidity index (THI); in dairy cows several THIs have been developed [20]. It is commonly accepted that cows experience mild heat stress when THI is between 68 and 74 and severe heat stress when cows are exposed to THI ≥75 (see review [21]). Although mild THI values have been associated with behavioural changes and lower reproductive performance (see review, [22]), reductions in milk production usually occur after 2 to 4 d of high THI [20,23]. Several management strategies have been developed to reduce cows’ body temperature and consequently, the negative impacts of heat stress. In indoor housed cows, the use of fans and air-conditioning are affective ways of reducing heat stress (see review [24]). For grazing cows, having access to shade reduced physiological and behavioural signs of heat stress [25]. Water consumption can also reduce cows’ body temperature [26], thus providing unrestricted access to water may mitigate the negative effect of heat stress on milk production of grazing cows.

The aim of this study was to measure the association between restricted and unrestricted access to water while on pasture on milk production of intensively managed grazing dairy herds.

## 2. Materials and Methods 

This project was approved by the Ethics Committees on Research in Humans (Protocol # PP1237779, 2015) and Animals (Protocol # PP00949, 2014) of the Universidade Federal de Santa Catarina and by the University of British Columbia Animal Care Committee (Protocol # A15-0082). Ruan R. Daros, José A. Bran, and two research assistants visited the farms and collected all the data. 

Between February and October of 2015, we visited 53 commercial dairy farms in the western part of Santa Catarina, Brazil (for a description of dairy systems in this region, see [3]). The grazing herds were enrolled as part of a larger study assessing risk factors for transition period diseases [27] and lameness [28], which required multiple visits (2–6) per farm. Detailed farm enrolment criteria is provided elsewhere [27]. In brief, intensively managed grazing herds (>16 h/d on pasture) with 40 to 100 cows were invited. Potential participants were recruited via telephone through referrals provided by people working in the local dairy sector. For this study we report only data collected in the first visit for every enrolled farm, which occurred between 27 February and 3 June 2015. 

On each farm, we undertook a semi-structured interview with the farmer, focusing on general herd management. Specifically, we included questions about the number of lactating and dry cows housed on each farm, number of fresh paddocks cows entered per day, hours spent in each paddock per day, and amount of concentrate offered per cow/d. We were not able to collect any concentrate samples in the current study. Farmers fed dairy cows corn silage throughout the year but were unable to provide an accurate response when asked to estimate the amount of corn silage fed daily. Therefore, farmers were asked to provide an estimate of the land area devoted to corn silage production, this information was then used as a proxy for amount of corn silage offered per cow/d. Farmers did not outsource corn silage, nor did they feed corn silage to any other livestock species kept on the farm. Daily herd milk production was reported by the farmer during the interview and cross-referenced with bulk tank milk shipment records. During the study period all farmers were paid according to the amount of milk produced instead of by unit of mass (e.g., kilograms of butterfat), and thus milk data were recorded as litres of milk.

After the interview the research team toured the farm and recorded the presence of water troughs in the grazing paddocks, along the walking lanes to and from the milking parlour and in the holding areas. Water troughs that were broken and unable to hold drinking water were recorded as not present. All water troughs were controlled by floating valves. Pasture was visually assessed in the grazing paddock that the lactating herd was present on visit date; pasture phenological stage and species were recorded (more details provided below). At each farm visit 1 full milking routine was observed, and number of milking cows and the breed of each cow recorded. All farms milked twice per day and followed a year-round calving schedule.

Hourly temperature, relative humidity and precipitation were retrieved from a regional meteorological station in São Miguel do Oeste, Santa Catarina (Ministério da Agricultura, Instituto Nacional de Meteorologia; elevation 665 m; 26°46′ S 53°30′ W). Participating farms were within a 50-km radius of the meteorological station.

### 2.1. Data Categorization, Handling, and Statistical Analysis

All data analysis and graphics were done in R version 3.5.0 [29]. Model assumptions were assessed graphically for normal distribution and homoscedasticity of the residuals. Multicollinearity among variables was assessed through variation inflation factor [30]. Raw data, code and output for the analysis are provided online as Appendix A.

#### 2.1.1. Outcome and Main Predictor

The outcome, average milk yield per cow/d per herd was calculated by the bulk milk shipped on the previous day divided by the number of milking cows at the moment of the visit. Our predictor of interest was type of water provision, which was divided in two categories: (1) Restricted access to water (*n* = 25)—cows had no access to water while on pasture, including during the night, and (2) Unrestricted access to water (*n* = 27)—cows either had at least one working water trough per grazing paddock or the paddock gate stayed open, allowing cows to voluntarily leave the paddock to gain access to water troughs. Water trough data was not collected for one participating farm. 

#### 2.1.2. Confounding Variables

To control for possible confounding of other parameters on the association of interest (effect of type of water provision on milk production), we fitted univariable linear regression models to measure associations between the outcome variable and possible confounding variables. The list of confounding variables was based on a casual diagram. Predictors were considered for inclusion in the final multivariable model if univariable model met assumptions criteria and if *p* < 0.2, as suggested by Dohoo et al. [30]. 

All cows were *Bos taurus* and were primarily Holsteins, Jerseys, and their cross. The effect of breed on milk yield was modelled using breed as a continuous predictor (% of Holstein cows in the herd) or as a two-level categorical predictor: >5% Holstein cows, (Holstein; *n* = 26 herds) or ≤75% Holstein cows, (non-Holstein; *n* = 27 herds). Only four herds were 100% Jersey, which prevented us from including a category for Jersey only. Modelling breed as a continuous predictor did not improve model fit; thus, the final model only included breed as a two-level categorical variable.

The amount of concentrate offered per cow/d was associated with milk production in the univariable model and was therefore kept in the final model.

Two variables regarding silage feeding were independently associated with the outcome variable: area of silage harvested per year/cow and number of silage feedings per day (categorized as 1, 2, or 3 times per day). However, number of silage feedings per day did not improve model fit and was therefore not included on the final model.

Pastures at the optimal cutting point should yield greater milk yield and that cows grazing pre-optimum cutting point pastures would rely less on free drinking water as juvenile pastures have greater moisture content. We therefore explored the variable pasture phenological stage, characterized into three levels (adapted from Moore et al. [31]) as a proxy for pasture DM content: before optimal cutting point (no presence of flowers and/or no senescent basal leaves), optimal cutting point (no presence of flowers and 1–3 senescent basal leaves), and after optimal cutting point (presence of flowers or >3 senescent basal leaves). To test if herds grazing forages at the optimum cutting point had higher milk production, we fitted a linear regression after dummy coding the variable pasture phenological stage (1 = optimal cutting point, 0 = other categories). To test if herds grazing forages at pre-optimum cutting point were less affected by water access type, we fitted a linear regression with the interaction term of type of water provision and pasture phenological stage (1 = pre-optimum cutting point, 0 = other categories), and their respective fixed effect. The variable phenological stage was not retained in the final model, as neither the effect of the interaction term between type of water provision and pasture phenological stage, nor its fixed effect were significant (see detailed description on Appendix A). These results will not be further reported.

Weather variables, relative humidity (%), and air temperature (˚F) were combined into THI following the formula described by Allen et al. [32]. Rolling average THI, number of hours of THI above 74, and cumulative precipitation (mm) of 4 d previous to visit date were calculated. Precipitation was further categorized into a two-level variable as “rain” (precipitation > 0 mm) or “dry” (precipitation = 0 mm).

Visit date was not considered for inclusion in the final model; however, visit date was ranked from first (27 February 2015) to last (3 June 2015) across farms, and a Wilcox Sum Rank test was used to investigate differences in visit dates between herds with different types of water provision.

#### 2.1.3. Final Model

Five herds that had incomplete data available were excluded, leaving 48 farms included in the final model. The final model was fitted using multivariable linear regression to measure the effect of type of water provision (unrestricted vs. restricted) on the average milk yield per cow/d. Breed (non-Holstein vs Holstein), amount of concentrate (kg/cow/d) and area of corn silage harvested (ha/cow/yr) were included as covariates in the model to control for possible confounding effects on the model parameter for water provision. Analysis of variance was performed to assess model fit, where the final model was compared with a model that was the same with the exception that the variable water provision was removed. The final model that included water provision decreased the residual sum of squares (i.e., explained more of the variation in the outcome variable) when compared to the model without the variable water provision. Based on these results we elected to retain the variable water provision in the final model.

## 3. Results

Descriptive data are presented as mean ± SD (for normally distributed variable) or median, minimum, 1st quartile (Q1), 3rd quartile (Q3), and maximum values (for non-normally distributed variables). Data from the final model (described below) are presented as mean, 95% confidence interval, and *p*-value.

### 3.1. Descriptive Results

Dairy herds averaged 37 (± 10) lactating cows. Except for 4 farms, all provided access to water troughs in the milking parlour holding area; all farms provided shade and water troughs in the non-grazing areas adjacent to the paddocks. All farmers reported bringing cows to these adjacent areas during the hot hours of the day (approximately 10:00 to 16:00 h). All farms used rotational grazing and all grazing paddocks were delimited by single electric wire. Twenty-six farms used fixed-size paddocks (i.e., the paddock fence was not moveable) while 20 farms used moveable fences to delineate paddock size. The remaining farms (*n* = 7) used fixed paddocks in some parts of the farm and moveable fences in others. The maximum time cows spent per paddock was 24 h/d. Farms varied in how often cows were provided access to fresh pasture each day: 8 provided fresh pasture once a day, 43 provided twice a day, and 1 farm did so three times per day. All farms offered corn silage to the lactating herd, either once per day (*n* = 12), twice per day (*n* = 30) or three times per day (*n* = 8). Deviations from 53 (i.e., total number of enrolled farms) in the sum of observations per variable are due to missing values. 

No association between visit date and type of water provision was found (W = 318; *p* = 0.73). Weather-related variables were not found to be associated with milk production and were not included in the final model. The descriptive statistics of the weather variables during the study period are presented on Table 1. 

### 3.2. Final Model

All variables used in the final model are presented in Table 2. Significant variables and their adjusted means based on the final model are presented in Figure 1 and Figure 2. Provision of unrestricted access to water was associated with a higher average milk yield per cow/d (1.7 L; 95% CI = 3.3–0.2; *p* = 0.03; Figure 1A) compared to restricted water access. This represents a difference of 10% in milk production between herds provided with unrestricted (adjusted mean: 18.1 L/cow/d) and restricted (adjusted mean: 16.4 L/cow/d) access to water. As expected there was an association between herd breed and milk production. Holstein herds produced on average 18% more milk per day per cow/d (2.8 L; 95% CI = 1.1–4.4; *p* < 0.01; Figure 1B) compared to non-Holstein herds (Holstein herds adjusted mean: 18.7 L/cow/d; non-Holstein herds adjusted mean: 15.9 L/cow/d). Amount of concentrate offered per cow/d was positively associated with average milk yield per cow/d (1.1 L increase in milk production/additional kg of concentrate offered/cow/d; CI = 0.6–1.6, *p* < 0.01; Figure 2). Area of silage/cow/yr was not significant (95% CI = −1.9–10.1, *p* = 0.18). The model explained 55% (adjusted R^2^) of the variation in average milk yield per cow/d.

## 4. Discussion

Though causal relationships cannot be inferred from observational studies, in this study herds that were provided unrestricted water access while grazing were associated with an approximately 10% higher milk production. In cattle, restricting water intake limits DM intake [18], which has experimentally been linked to reduced milk yield [33]. A recent meta-analysis on experiments comparing difference in milk yield of cows offered water access twice a day versus providing ad libitum water access found that restricting access to water was associated with reduced milk production [13]. Interestingly, these authors found an overall 1.7% reduction on milk yield in studies done in temperate climates, compared to a reduction of 15% on milk yield in experiments carried out in tropical regions. Some caution is warranted when interpreting their findings for tropical regions given that only two experiments using mainly *Bos indicus* cows (both with low animal numbers) met the inclusion criteria [13]. Our study, undertaken in a sub-tropical region, shows a reduction on milk yield in *Bos taurus* herds that had restricted access to water while grazing; the reduction was more dramatic than that shown for herds housed under cooler climates [13] and more in line with the results reported for tropical regions (e.g., [34]). It is worth noting that cows in herds with restricted water access had access to water troughs at least twice a day either before or after milking and that in hot days farmers brought the cows to areas with free access to shade and water troughs during the day.

We measured the association between pasture phenological stage and water provision type on milk production of the studied herds. Considering that juvenile plants have higher moisture content, the milk production of cows grazing such pastures, perhaps, would not be as negatively impacted by water provision type as total daily water intake would depend less on free drinking water intake. However, when grass phenological stage was tested in the univariable model there was no effect of this variable on milk production. Alternatively, farmers could be adjusting the levels of concentrate and corn silage based on pasture conditions, resulting in no differences in milk production across farms with different pasture conditions. Another confounding factor that could have influenced our results was that we collected our data during the time span of two seasons (summer and fall), thus weather-related variables and changes in day light length may have affected our results. When tested in the univariable model, the effect of daily average THI or number of hours THI was above 74 in the previous 4 days before visit day did not indicate an association between these variables and milk production. Differently from reproductive performance and behavioural changes [21,32], milk production is usually negatively impacted after periods of THI higher than 74 [23]. Perhaps we did not find an association between THI and milk production because THI was not extreme during the study days (see Table 1). 

Lack of, or poor access to, water in grazing dairy herds is common in the study region [11] and elsewhere in the world (e.g., Europe; [9,35]), Restricted water access for lactating cows housed on pasture areas has been previously associated with lower animal welfare standards [9]. The association between restricting access to water and animal welfare is obvious, as thirst has been long argued to be a negative emotional state [10]. Indeed, animal welfare assurance programs such as the Canadian Code of Practice for the Care and Handling of Dairy Cattle [36], the FARM (Farmers Assuring Responsible Management) program in the United States of America [37] and Welfare Quality^®^ in Europe [38] all state that cattle should have access to water to meet their needs and avoid prolonged periods of thirst. Our work is limited in that we could not assess individual data on time spent without access to water and other measures of thirstiness. We therefore encourage future research in this area, especially in the development of methodologies to assess thirst in cattle.

In cows that have unrestricted access to water, drinking behaviour is most often synchronized with milking [39]. Although we did not measure cows’ drinking behaviour in the present study, we speculate that competition to gain access the water troughs at milking times, and during elevated temperatures, was likely high (see e.g., [25]). Limiting access to water could result in some cows experiencing thirst-like states. In the same study region, farmers have reported bringing their cows to shade during hot hours of the day [11], and indeed in our study all farmers reported employing a similar practice, bringing the cows to a separate area with access to shade and water troughs. However, we caution using this strategy as it may not mitigate heat stress entirely, given that the farmers are making the decisions on behalf of the cows as to when they need access to shade and water; cows experience hot temperature differently than humans (see review [22]). Based on our result, this practice is not sufficient to mitigate the negative effects of restricted water access on milk production. 

Diet content is known to affect milk yield (see e.g., [40,41]); we noted a positive relation between amount of concentrate offered per cow/d and daily herd milk production, similar to that found in a previous study on grazing dairy cattle [41]. Although the average concentrate provided to cows varied across farm from a minimum of 1.5 to a maximum of 9 kg/cow/d, we have purposely not provided any recommendations regarding daily intakes of concentrate given that forage quality is an important factor that should also be considered; we did not collect any information on concentration composition or forage quality. We also recognize that the variables used to assess amount of supplementation eaten by the cows (amount of concentrate offered and area of silage harvested/cow/yr) are only estimates, but our approach did allow us to control for possible confounding effects regarding the association between water access and milk yield. In the same study area, recent research has showed that area of corn silage produced was positively associated with herd milk production [42].

In conclusion, providing unrestricted access to water while grazing was associated with higher milk production, regardless of the breed of cows and the amount of concentrate and silage offered. Water availability should be considered when planning dairy management systems, as restricted water access may decrease milk production.

## Figures and Tables

**Figure 1 animals-09-00048-f001:**
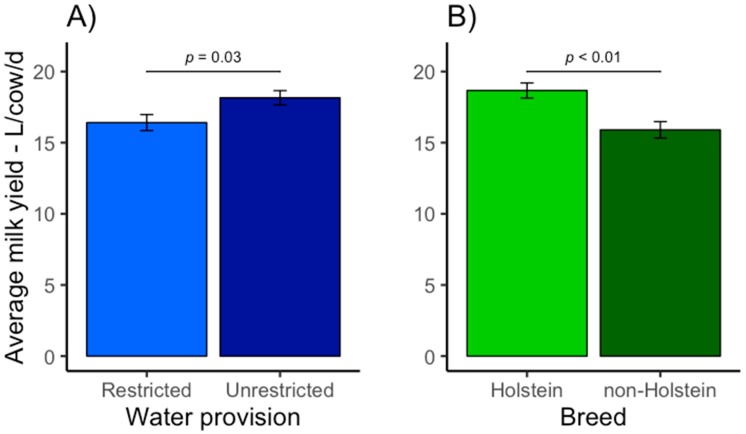
The effects of water provision (**A**) and breed (**B**) on daily milk production (L/cow/d) on grazing dairy herds (*n* = 48) in Santa Catarina State, Brazil. Means are presented as adjusted means and standard errors.

**Figure 2 animals-09-00048-f002:**
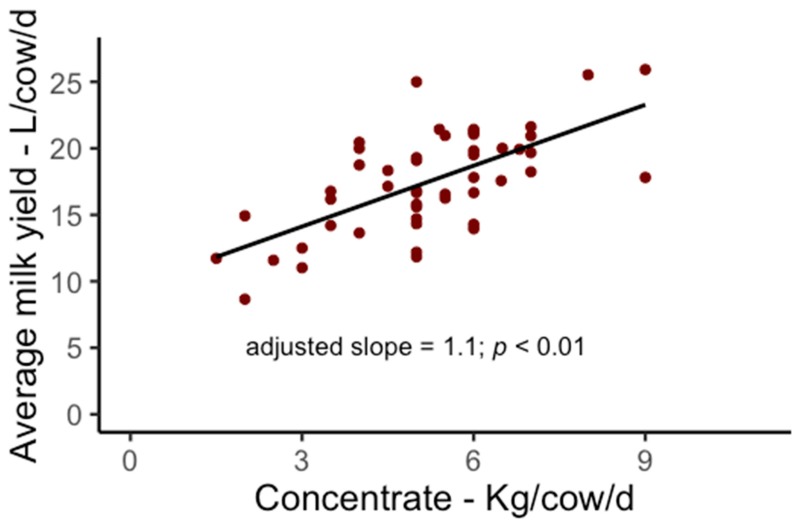
The effects of daily concentrate allocation per cow (kg/cow/d) on daily milk production (L/cow/d) on grazing dairy herds (*n* = 48) in Santa Catarina State, Brazil. Dots represent the raw values summarizing the average of each farm.

**Table 1 animals-09-00048-t001:** Description of weather data during the study period.

Variable	min	Q1 ^1^	median	Q3 ^1^	max
Daily air temperature (˚C)					
Average	11.4	17.7	19.6	23.8	26.0
Average min	7.6	13.8	17.6	19.4	22.3
Average max	13.4	21.6	24.4	29.0	30.2
Daily Relative humidity (%)					
Average	57.7	67.5	77.9	85.0	97.6
Average min	34.0	50.5	56.5	66.0	94.0
Average max	71.0	86.0	96.5	98.8	100
Daily THI ^2^					
Average	52.3	63.0	66.3	71.2	74.9
Average min	45.8	56.9	63.2	66.3	70.5
Average max	56.4	69.0	71.7	76.9	79.3
THI 4-d rolling average ^3^					
Average	55.3	63.7	68.2	71.3	72.9
Sum of hours THI >74	0	0	8	27	34
Precipitation (mm/d)	0	0	0.2	1.8	18.6

^1^ 1st quartile (Q1); 3rd quartile (Q3). ^2^ Temperature humidity index. Calculated as described by Allen et al. [32]. ^3^ Average temperature humidity index (THI) and sum of hours of THI >74 during the 4 days prior to visit date.

**Table 2 animals-09-00048-t002:** Description of variables included in the final model ^1^.

Variable	Type	*n* ^2^	Mean	SD	Median	Min	Max
Herd average milk yield (kg/cow/d)	continuous	51	17.7	4.0	17.8	8.7	27.1
Silage area -hectare/cow/yr	continuous	51	0.27	0.13	0.24	0.10	0.77
Concentrate -kg/cow/d	continuous	50	5.2	1.6	5.0	1.5	9.0
Breed ^3^ *Holstein* *non-Holstein*	categorical	53 *26* *27*					
Water provision ^4^ *Restricted* *Unrestricted*	categorical	52 *24* *27*					

^1^ In total five herds had incomplete data, thus only 48 herds were included in the model. ^2^ Deviations from 53 in the sum of observations per variable is due to missing values. ^3^ Holstein (herds >75% Holstein cows); non-Holstein (herds ≤ 75% Holsteins). ^4^ Restricted: paddocks without water trough and gate stayed closed. Unrestricted: herds that had one water through per grazing paddock and/or paddock gate stayed open. Italics and indentation reflect the levels of the categorical variable.

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
