# Peer review of "Readily Available Water Access is Associated with Greater Milk Production in Grazing Dairy Herds"

_animals, 2019, doi:10.3390/ani9020048_

Round 1

Reviewer 1 Report

By far the best manuscript I have ever reviewed. The topic is of high importance and the article is well-structured and well written. I only have a few minor comments:

In section 2.1.2 the different confounding variables are described. However, I miss an equivalent description of the concentrate variable. I see in table 2 that this is a cont. variable. Did farmers use different types of concentrate? Did e.g. DM differ?

In section 3.2 you say that more concentrate = more milk. Is there a cutoff point? Is it possible to suggest an optimal amount of concentrate based on existing literature?

In section 3.2, would it be possible to provide the difference in percent in addition to the specified liters (1.7 and 2.8). If – for ease – we imagine a Holstein cow produces twice that of a Jersey, I think this difference would be less dramatic and more informative in percent for the Holstein vs non-Holstein herds. Providing the average national milk yield of the breeds might be interesting.

Line 163: There is an asterisk, but it is unclear that this refers to.

Line 290: “Based on our results, …”?

Author Response

Response to Reviewer 1 Comments

By far the best manuscript I have ever reviewed. The topic is of high importance and the article is well-structured and well written. I only have a few minor comments:

Authors: Thank you very much for your timely review and kind words. Below we have responded to each of your requests.

In section 2.1.2 the different confounding variables are described. However, I miss an equivalent description of the concentrate variable. I see in table 2 that this is a cont. variable. Did farmers use different types of concentrate? Did e.g. DM differ?

Authors: We have added the description requested on lines 148 -149 to read as follows: “Amount of concentrate offered per cow/d was associated with milk production in the univariable model and was therefore kept in the final model.“

We have added a sentence to the methods section (lines 101 – 102) to read as follows: “We were not able to collect any concentrate samples in the current study.”

In section 3.2 you say that more concentrate = more milk. Is there a cutoff point? Is it possible to suggest an optimal amount of concentrate based on existing literature?

Authors: In pasture-based systems, forage quality can have a profound effect on forage intake, and thus milk response to concentrate feeding is likely also highly variable under these conditions.  We therefore are unable to provide any recommendations regarding recommended daily concentrate intakes for lactating cows housed under these types of grazing conditions. We have now included a sentence (Lines 305 – 309) in the discussion to avoid any extrapolation of our findings. Lines 305 – 309 reads: “Although the average concentrate provided to cows varied across farm from a minimum of 1.5 to a maximum of 9 kg/cow/d we have purposely not provided any recommendations regarding daily intakes of concentrate given that forage quality is an important factor that should also be considered; we did collect any information on concentration composition or forage quality.

In section 3.2, would it be possible to provide the difference in percent in addition to the specified liters (1.7 and 2.8). If – for ease – we imagine a Holstein cow produces twice that of a Jersey, I think this difference would be less dramatic and more informative in percent for the Holstein vs non-Holstein herds. Providing the average national milk yield of the breeds might be interesting.

Authors: We now include this information as requested on Lines 216 - 220: “Provision of unrestricted access to water was associated with a higher average milk yield per cow/d (1.7 L; 95% CI = 3.3 – 0.2; P = 0.03; Fig. 1 A) compared to restricted water access. This represents a difference of 10% in milk production between herds provided with unrestricted (adjusted mean: 18.1L/cow/d) and restricted (adjusted mean: 16.4L/cow/d) access to water.”

Lines 220 – 223 now reads: “As expected there was an association between herd breed and milk production. Holstein herds produced on average 18% more milk per day per cow/d (2.8 L; 95% CI = 1.1 – 4.4; P < 0.01; Fig. 1 B) compared to non-Holstein herds (Holstein herds adjusted mean: 18.7 L/cow/d; non-Holstein herds adjusted mean: 15.9 L/cow/d)”

Line 163: There is an asterisk, but it is unclear that this refers to.

Authors: the asterisk indicated the interaction term between the two variables. We have now replaced the asterisk with the word “and”, now on line 164.

Line 290: “Based on our results, …”?

Authors: Fixed. Now on line 300. Thank you!

Reviewer 2 Report

General comments.

The manuscript has been written to a good standard. However, I had just one issue to raise on how the results have been interpreted and concluded as this caught my attention. 

While the P value might be significant, it doesn't prove causality. The authors should interpret the results based on the Confidence intervals / and or Odds ratios. I suggest the results are tapered down and odds ratio and (C.I) used to make the conclusions. I wouldn't rely on the p values.

Author Response

Response to Reviewer 2 Comments

General comments.

The manuscript has been written to a good standard. However, I had just one issue to raise on how the results have been interpreted and concluded as this caught my attention.  

While the P value might be significant, it doesn't prove causality. The authors should interpret the results based on the Confidence intervals / and or Odds ratios. I suggest the results are tapered down and odds ratio and (C.I) used to make the conclusions. I wouldn't rely on the p values. 

            Authors: Thank you for your comments. We agree that in such observational studies we cannot infer causality. We have now fixed this issue throughout the manuscript. See below.

            End of abstract:

Lines 33 - 34, now reads: “In conclusion, providing free access to drinking water while grazing was associated with greater milk production”

            Results:

            Lines 216 – 217, now reads: “Provision of unrestricted access to water was associated with a higher average milk yield per cow/d…”

Lines 220 – 221, now reads: “As expected there was an association between herd breed and milk production. Holstein herds …”

Lines 223 – 224, now reads: “Amount of concentrate offered per cow/d was positively associated with average milk yield per cow/d (1.1 L increase in milk production/additional kg of concentrate offered/cow/d; CI = 0.6 – 1.6, P < 0.01; Fig. 2).”

Conclusion:

Lines 315 - 318. Our conclusion now reads: “In conclusion, providing unrestricted access to water while grazing was associated with higher milk production, regardless of the breed of cows and the amount of concentrate and silage offered. Water availability should be considered when planning dairy management systems, as restricted water access may decrease milk production and can impair animal welfare.”

Regarding confidence intervals and odds ratio we have elected to retain the confidence intervals given that our outcome variable is continuous; see results presented on Lines 217 – 218; 222; 224 – 225 and 226). We believe that the changes we made throughout the manuscript now reflect the analysis and the results we have reported.

We also added more details on our hypothesis testing on section 2.1.3. Lines 183 to 188 reads: “Analysis of variance was performed to assess model fit, where the final model was compared with a model that was the same with the exception that the variable water provision was removed. The final model that included water provision decreased the residual sum of squares (i.e. explained more of the variation in the outcome variable) when compared to the model without the variable water provision. Based on these results we elected to retain the variable water provision in the final model.”

Thank you very much for your comments and the timely review.